# The Divergent Roles of the Rice bcl-2 Associated Athanogene (BAG) Genes in Plant Development and Environmental Responses

**DOI:** 10.3390/plants10102169

**Published:** 2021-10-13

**Authors:** Hailian Zhou, Jiaying Li, Xueyuan Liu, Xiaoshuang Wei, Ziwei He, Lihua Hu, Jibin Wang, Mingzheng Duan, Guosheng Xie, Jihong Wang, Lingqiang Wang

**Affiliations:** 1State Key Laboratory for Conservation & Utilization of Subtropical Agro-Bioresources, College of Agriculture, Guangxi University, Nanning 530004, China; 2017401011@st.gxu.edu.cn (H.Z.); 1917301040@st.gxu.edu.cn (X.W.); 1931200514@st.gxu.edu.cn (Z.H.); hulihua@gxu.edu.cn (L.H.); wangjibin1988@sina.cn (J.W.); duanmingzheng@hotmail.com (M.D.); 2College of Plant Science and Technology, Huazhong Agricultural University, Wuhan 430070, China; jiayingli@webmail.hzau.edu.cn (J.L.); xueyuan_liu1995@163.com (X.L.); xiegsh@mail.beau.edu.cn (G.X.); 3Department of Life Science, Tangshan Normal University, Tangshan 063000, China

**Keywords:** rice, BAG, cell wall, cellulose synthase

## Abstract

Bcl-2-associated athanogene (BAG), a group of proteins evolutionarily conserved and functioned as co-chaperones in plants and animals, is involved in various cell activities and diverse physiological processes. However, the biological functions of this gene family in rice are largely unknown. In this study, we identified a total of six BAG members in rice. These genes were classified into two groups, *OsBAG1*, *-2*, *-3*, and *-4* are in group I with a conserved ubiquitin-like structure and *OsBAG5* and *-6* are in group Ⅱ with a calmodulin-binding domain, in addition to a common BAG domain. The *BAG* genes exhibited diverse expression patterns, with *OsBAG4* showing the highest expression level, followed by *OsBAG1* and *OsBAG3*, and *OsBAG6* preferentially expressed in the panicle, endosperm, and calli. The co-expression analysis and the hierarchical cluster analysis indicated that the *OsBAG1* and *OsBAG3* were co-expressed with primary cell wall-biosynthesizing genes, *OsBAG4* was co-expressed with phytohormone and transcriptional factors, and *OsBAG6* was co-expressed with disease and shock-associated genes. β-glucuronidase (GUS) staining further indicated that *OsBAG3* is mainly involved in primary young tissues under both primary and secondary growth. In addition, the expression of the *BAG* genes under brown planthopper (BPH) feeding, N, P, and K deficiency, heat, drought and plant hormones treatments was investigated. Our results clearly showed that *OsBAGs* are multifunctional molecules as inferred by their protein structures, subcellular localizations, and expression profiles. *BAGs* in group I are mainly involved in plant development, whereas *BAGs* in group II are reactive in gene regulations and stress responses. Our results provide a solid basis for the further elucidation of the biological functions of plant *BAG* genes.

## 1. Introduction

The Bcl-2-associated athanogene (BAG) family is evolutionarily conserved and function as co-chaperones [1,2,3]. The first *BAG* gene was discovered in a screen of a small Mus musculus embryo cDNA library using a recombinant human Bcl-2 protein as a bait to identify b-cell lymphoma 2 (Bcl-2) interactors and is named as *BAG1* [4]. *BAG1* can enhance cell survival synergistically with Bcl-2, representing a novel type of anticell death gene in apoptotic PCD pathways. BAG proteins are distinguished by a common conserved region located near the C-terminal, termed the BAG domain (BD), which binds to the ATPase domain of Hsp70/Hsc70 molecular chaperones while the N-terminal binds to multiple molecular chaperones, forming a ternary complex and playing the “bridge role” of molecular chaperone [5,6]. BAG proteins have been shown in animals as functioning in various cell activities and diverse physiological processes such as division, migration, apoptosis, autoimmunity, tumorigenesis, neuronal differentiation, stress responses, and the cell cycle [2,5,6,7].

*BAG* genes are evolutionarily conserved in plants [2,8]. With bioinformatic approaches, BAG proteins are successively identified in *Arabidopsis thaliana* (*A. thaliana*), wheat (*Triticum*
*aestivum* L.), rice (*Oryza sativa* L.), soybean (*Glycine max*), *Gossypium raimondii*, *Solanum lycopersicum*, and grape (*Vitis vinifera*) [2,3,9]. In *Arabidopsis,* seven homologs of the BAG family are identified. Among them, four members have a similar domain organization to that of animal BAG proteins, whereas the other three contain a calmodulin-binding domain possibly reflecting the differences in their functions between plants and animals [2]. The individual members of the BAG family have diverse subcellular localizations and can be located in the endoplasmic reticulum (ER), mitochondria, and tonoplast [3,10,11,12]. Plant BAG proteins are multifunctional, as they inhibit several cell-death-mediated processes and appear to function in cyto-protection under both stress conditions and plant development [2,3,10,13,14]. In *Arabidopsis*, *AtBAG5* was found not only to regulate the production of reactive oxygen species (ROS), but also to mediate the aging of leaves by regulating the expression of senescence-associated genes (SAGs) [11]. The *atbag6* mutants show enhanced susceptibility to the necrotrophic fungus Botrytis cinerea, and the protein was confirmed to interact with bag-associated GRAM protein (BAGP1) and aspartyl protease-cleaving BAG (APCB1) separately to form a complex and jointly participate in the resistance to fungal infection [2,15]. It was also shown that ethylene and salinity antagonistically control the transcription of *BAG6* and *BAG7* genes to mediate the salt-induced cell death [16]. Transgenic banana plants overexpressing *MusaBAG1* exhibit enhanced resistance to Fusarium wilt [17]. A grape Bcl-2-associated athanogenea (*HSG1*) is not expressed under normal conditions, but the expression level in leaves and fruits increases sharply under heat stress and promotes floral transition by activating CONSTANS expression in transgenic *Arabidopsis* plant [18]. A plenty of evidence proofed that plant BAG proteins have a similar modulating mechanism to that of animal BAGs, and they can interact with other proteins as molecular bridges to mediate diverse cell events.

Although BAG proteins are being extensively studied in dicotyledons plant *A. thaliana* [2,3,15,19], little has been studied in other plants. Rice is one of the major food crops worldwide and serves as a model plant for the functional genomic characterization of the monocotyledon. Using bioinformatic approaches, a previous study identified six BAG protein homologs in the rice genome sequence [20]. They reported the genome organization, phylogeny, and the gene expression analysis of *OsBAG* multigene family in rice, which has laid a foundation for further functional studies. Recently, *OsBAG4* protein was shown to interacted with RING-type E3, which is encoded by the enhanced blight and blast resistance 1 (gene ebr1), leading to its ubiquitination and degradation. *OsBAG4* accumulation in *ebr1* or *OsBAG4* overexpression triggers autoimmunity and broad-spectrum disease resistance and, most recently, was reported to be involved into the salt tolerance [6,21]. However, so far, the function of other *BAGs* in rice has not been known yet. As shown by the previous studies in *A. thaliana*, the functions of the different members in the BAG family could be quite different from each other. For examples, *AtBAG4* is involved in response to the UV light, salt, and drought, whereas the *BAG5* is in response to Dark, *BAG6* is in response to Fungal and *BAG7* is in response to the stimulation of heat and cold [2,3,10,11,14,19,22,23,24]. Therefore, the elucidation of the localization and functions of the other protein members of the rice BAG family under various cell states and development stages is of interest. In this study, we reported the functional classification of the genes from the BAG family in rice. We first described genome organization and performed a phylogenetic and structural analysis to determine potential functions of rice *BAGs*. Then, we focused on an integrated expression analysis of the *BAG* genes by pooling information from Collections of Rice Expression Profiling (CREP) whole growth period microarray databases (http://crep.ncpgr.cn, accessed on 18 April 2021) and from several other databases (RiceXPro; https://ricexpro.dna.affrc.go.jp/, accessed on 18 April 2021; https://www.ncbi.nlm.nih.gov/, accessed on 18 April 2021). Finally, the vector of the β-glucuronidase (GUS) report gene was employed to construct the *BAG* promoter-driven expression plasmid, and the recombinant plasmid was introduced to the rice plants to verify the *BAG* gene we are interested in. Our results implied that the rice *BAG* family members are multifunctional to development and stresses.

## 2. Materials and Methods

### 2.1. Database Searches for OsBAG Genes in Rice

The hidden Markov model (HMM) profile of the BD (PF02179) was downloaded from PFam (http://pfam.sanger.ac.uk/, accessed on 18 April 2021). We employed a name search and the protein family ID PF02179 for the identification of *OsBAG* genes from the rice genome. Information about the chromosomal localization, coding sequence (CDS), amino acid (AA), and full-length cDNA accessions was obtained from TIGR (http://www.tigr.org, accessed on 18 April 2021) and KOME (http://cdna01.dna.affrc.go.jp/cDNA, accessed on 18 April 2021). The corresponding protein sequences were confirmed by the Pfam database (http://www.sanger.ac.uk/Software/Pfam/search.shtml, accessed on 18 April 2021).

### 2.2. Sequence and Structure Analysis

We performed our exon–intron structure analysis using GSDS (http://gsds.cbi.pku.edu.cn/, accessed on 18 April 2021) [25]. The protein transmembrane helices were predicted by the TMHMM Server V2.0 (http://www.cbs.dtu.dk/services/TMHMM/, accessed on 18 April 2021). The theoretical isoelectric point (pI) and molecular weight (Mw) of protein were caculated by ExPASy Server (https://web.expasy.org/compute_pi/, accessed on 18 April 2021). Protein subcellular locations were analyzed using a comprehensive predictor CropPAL, which was used to calculate and summarize six predictors and achieve a more accurate result by consensus algorithm SUBAcon (http://croppal.org/, accessed on 18 April 2021) [26].

### 2.3. Phylogenetic Analyses and Motif Identification

The alignment of rice and *A. thaliana* BAG protein sequences as well as the rice BD were performed with ClustalX, and the unrooted phylogenetic trees were constructed with the MEGA7.0 program and the neighbor-joining method with the Poisson model and 1000 bootstrap replicates [27]. Protein sequences were analyzed using the MEME program (http://meme.sdsc.edu/meme/cgi-bin/meme.cgi, accessed on 18 April 2021) for the confirmation of the motifs. The MEME program (version 5.1) was employed with the following parameters: number of repetitions, any; maximum number of motifs, 10; optimum motif width set, >6 and <200.

### 2.4. Co-Expression Analysis of OsBAGs and OsCESAs in Rice

The expression profile data of the *OsBAG1*, *3*, *4*, *6* genes and the *OsCESAs* in 33 tissue samples (Appendix A) of Zhenshan 97 (ZS97) and Minghui 63 (MH63) were obtained from the CREP database maintained in Huazhong Agricultural University, Wuhan 430070, China (http://crep.ncpgr.cn, accessed on 18 April 2021), which was generated by a rice transcriptome project using the Affymetrix Rice GeneChip microarray. This array contains probes to query 51,279 transcripts representing two rice cultivars, with approximately 48,564 *japonica* transcripts and 1260 transcripts representing the *indica* cultivar. This unique design was created within the Affymetrix GeneChip Consortia Program and provides scientists with a single array that can be used for the study of rice. High-quality sequence data were derived from GenBank mRNAs, TIGR gene predictions, and the International Rice Genome sequencing project. The arrays were designed using NCBI UniGene Build #52, incorporating predicted genes from GenBank and the TIGR Os1 v2 dataset. Background correction and quantile normalization were implemented using the robust multi-array average method. The intensities of perfectly matched probes were extracted. Probes with a hybridization intensity in Zhenshan 97 or Minghui 63 of <8.4 (the 0.1 quantile of minimum probe intensity) were masked as low-quality probes, and a customized chip description library (R package) with unmasked probes was generated and used in the analyses. The probe intensity files (.cel files) resulting from RNA hybridization were read into R. Background correction, and quantile normalization and summarization were also performed using the robust multiarray average method in Bioconductor Affy package. More details of transcriptomic experiments using Affymetrix Gene Chip can be found in a previous study [28]. The expression data of *OsBAG2* and -*5* in the Nipponbare were downloaded from RiceXPro database (https://ricexpro.dna.affrc.go.jp/, accessed on 18 April 2021) (Appendix A).

The expression profile data of the top 100 correlated genes with the *OsBAGs* genes were browsed in an Affymetrix Rice GeneChip microarray. The correlation coefficients of these genes with the *OsBAGs* were calculated, and the data in Excel file format were put into UCINET 6.0 to conduct the network analysis. Then, the network adjacency matrix files were visualized with NetDraw. In the CREP study, the tissues and organs were collected throughout the life cycle of the rice, and the total number was 33 (*n* = 33). Thus, we queried the correlation coefficient significance checklist and found the r threshold values of this analysis (free degree was 32) were 0.339 (*p* = 0.05) and 0.539 (*p* = 0.01). Since the higher strength of the linear association in a co-expression analysis should be better to assure that obtained results are biologically relevant as suggested by a previous study [29], we understand that the strength of the linear association was also affected by the heterogeneity degree between the samples investigated and the expression pattern of gene pairs themselves. Therefore, in this study, we chose 0.6 which is above the threshold of 0.523 (*p* = 0.01) in this study, that is, Pearson correlation coefficient of showing lines are >0.6.

### 2.5. Expression Profile of OsBAGs in Responses to Environmental Stimulations

To detect the *OsBAGs* expression in response to the brown planthopper (BPH), 14-day-old seedlings of the varieties 9311 were treated with eight insects (third-instar) per plant and sampled at 0, 24, and 48 h after the BPH release. Total RNAs were isolated from the tissues at different time points as indicated using a Hipure plant RNA Mini Kit. IMPLEN P300 was used for RNA quantification. Reverse transcription-PCR (RT-PCR) was conducted with total RNA according to the manufacturer’s instructions of the PrimeScriptTM RT reagent kit with gDNA Eraser (TaKaRa, RR047A). qPCR was performed with the Analytik jena qTOWER3 Real-Time qPCR system using the Taq Pro Universal SYBR qPCR Master Mix (Vazyme, Q712) with the following PCR conditions: 95 °C for 30 s, 95 °C for 10 s for 40 cycles, and 60 °C for 20 s. Relative gene-expression levels were calculated using the 2^–ΔΔCT^ method and normalized with the gene *Osactin* [30]. Gene primers for qRT-PCR were designed according to the reference cDNA sequences of Nipponbare (Appendix A) [31].

The expression profile data of *OsBAGs* in root from rice seedlings under nitrogen (N), phosphorus (P), and potassium (K) deficiency conditions were obtained from the RiceXPro database (https://ricexpro.dna.affrc.go.jp/, accessed on 18 April 2021). The 7-day-old seedlings were exposed to N, P, and K deficiency treatments and control conditions separately. Root samples were collected at 6 and 24 h after the treatments. The expression profile data of *OsBAGs* in heat and drought stimulations were obtained from the NCBI GEO database (https://www.ncbi.nlm.nih.gov/, accessed on 18 April 2021), and the respective GEO accessions were GSE83542 and GSE41647. A comparative analysis of transcriptome profiles in panicles from two rice lines, heat-tolerant line 252 (HTL252) and heat-susceptible line 082 (HSL082), was performed by using rice Affymetrix GeneChip. After a continuous high-temperature treatment for 5 d from both lines and each treatment type, the panicles were collected for RNA extraction and hybridization on Affymetrix microarrays. For each treatment, three biological replicates were carried out. The transcriptomes of two contrasting varieties, Dagad deshi (tolerant) and IR20 (susceptible), under both control and stress conditions, were analyzed with the Affymetrix microarray platform, to elucidate the differences in their responses to drought stress. Hydroponically grown seven-day-old seedlings of the two varieties were subjected to drought stress by placing them on 3 mm Whatmann sheets under light at 28 ± 1 °C for 3 and 6 h. For the control samples, seedlings were kept in a root growth medium at 28 ± 1 °C for 3 and 6 h [32,33].

The expression profile data of *OsBAGs* in root and shoot from rice seedlings treated with six plant hormones, namely abscisic acid (ABA), gibberellic acid (GA3), indole-3-acetic acid (IAA), brassinolide (BL), trans-zeatin (tZ), and jasmonic acid (JA), were obtained from the RiceXPro database (https://ricexpro.dna.affrc.go.jp/, accessed on 18 April 2021). Seeds of japonica rice cultivar Nipponbare were germinated and grown hydroponically in a growth chamber at 28 °C under continuous light. Seven-day-old seedlings were transferred in a culture solution containing the hormone and in a culture solution without hormone to serve as a control (mock treatment). The samples were collected after 1 and 3 h incubation for root with three replicates and after 1 and 3 h incubation for shoot with two replicates. Cy3 (mock treatment) and Cy5 (hormone treatment) were used for hybridization using the Agilent two-color microarray analysis system. The time-course expression profile for each gene was shown as the log-ratio of signal intensity (log2 Cy5/Cy3).

### 2.6. GO/KEGG Pathway Enrichment

The Gene Ontology (GO) and KEGG pathway enrichment was analyzed by using DAVID 6.8 (https://david.ncifcrf.gov/tools.jsp, accessed on 18 April 2021) and KOBAS 3.0 (http://kobas.cbi.pku.edu.cn/kobas3, accessed on 18 April 2021) [34,35].

### 2.7. Plasmid Construction and Plant Transformation

To fuse the *LOC_Os06g03640* promoter to the GUS gene, the promoter of *OsBAG3*, a 1731bp fragment upstream of the ATG of *LOC_Os06g03640*, was amplified by PCR. F primer TCATGCTGCCTGCTGACCTA and R primer CATCCTCCTCTCCTCCTCTTCTTCTC were searched by Primer5.0 and blasted for specificity in the NCBI. Rice genomic DNA used as the template for the PCR was extracted using the CTAB method as reported in a previous study [36]. The PCR reaction mixture was prepared as follows: 1.0 μL DNA template, 4.0 μL 10× buffer, 1.0 μL dNTPs (10 mM), 1.0 μL forward primer (10 μM), 1.0 μL reverse primer (10 μM), and 0.4 μL blend taq (1 U/μL), Then, ddH_2_O was added to the PCR reaction mixture with a total volume of 40 μL. The whole PCR procedure included three steps: a denaturation at 95 °C for 5 min first, PCR cycling (32 cycles) consisting of a denaturation step at 95 °C for 30 s followed by annealing at 58 °C for 30 s and extension at 72 °C for 2 min, and a final extension at 72 °C for 10 min. The amplified PCR product was subjected to agarose gel electrophoresis (2% concentration) and cloned to pEasy-T3. The correct clone was introduced into the GUS fusion vector pC130gT to produce ProOsBAG3::GUS after sequencing. The construct was then introduced into Agrobacterium tumefaciens EHA105 and was transformed into the callus derived from japonica cultivar Nipponbare by Agrobacterium-mediated transformation, as previously described by Wu et al. [37].

### 2.8. GUS Staining

The histochemical analysis of the GUS reporter enzyme was performed essentially according to the method described by a previous study [38]. Sample tissues were incubated in a reaction buffer for 2 d, and the GUS staining pattern was observed under a stereomicroscope (Leica S8AP0, Weztlar, Germany).

## 3. Results

### 3.1. The Identification of the OsBAG Family in Rice

Searching the TIGR database revealed six sequences that significantly matched to the BAG protein family in rice, and all of them contained conserved a BD (Table 1). The sequence length of *BAG* genes varied from 642 to 1368 bp (214 to 456 AA), with the *OsBAG5* being much shorter than the others and appearing to be truncated. In addition to the BD, the OsBAG1–4 contained a ubiquitin-like (UBL) structure, whereas OsBAG5 and -6 harbored a calmodulin-binding domain, a novel feature only found in plant BAG proteins, indicating possible divergent mechanisms associated with plant-specific functions (Table 1; Figure 1A). In addition, all genes had KOME cDNA support, and probes for four of them (*OsBAG1*, *-3*, *-4*, and *-6*) could be found in the CREP database (Table 1). In our study, the protein subcellular locations were analyzed using a comprehensive predictor CropPAL, which was used to calculate and summarize six predictors, and the accurate results can be achieved by the consensus algorithm SUBAcon (http://croppal.org/, accessed on 18 April 2021) [26]. Here, we just presented the result of “Winner Takes All”, which integrated the results from the all six predictors. *OsBAG2*, *-4*, and *-5* had a predicted cytosolic localization and OsBAG3 and -6 were predicted to localize in the nucleus, whereas OsBAG1 had a predicted mitochondrial localization (Table 1). However, the *OsBAG2*, *-4*, and *-5* also had a high possibility in the nucleus. We noticed that the locations of *OsBAG1* and *-6* predicted in this study were different with those in a previous study, which used the software WoLF PSORT [20]. For comparison, we listed all the predication results in Appendix A. We can see that basically our predication is the same as that of the study conducted by Rana et al. where the same software WoLF PSORT was used [20]. Further studies are needed to confirm the subcellular locations of the BAG proteins. The *OsBAG* gene members were distributed on six chromosomes of rice individually, and no tandem duplication set was discovered, which was different with the *Arabidopsis* that had four *BAGs* on chromosome V likely arising from local gene duplication (Appendix A).

### 3.2. Structural and Phylogenetic Analyses of OsBAGs

An unrooted phylogenetic tree was generated from the alignments of the six protein sequences with two distinct clusters (Figure 1B). When referring to the BAG classification in *Arabidopsis*, the six proteins could be divided into two groups (Figure 1B). Group I contained OsBAG1, -2, -3, and -4, while group II contained OsBAG5 and -6. This pattern of clustering could possibly reflect the related functions of OsBAG proteins within each group. Our result was consistent with that of a previous study [20].

The members in the same groups tended to share a similar exon–intron organization, motif composition, length of cDNA, and PI values of the protein, especially among *OsBAG1*, *-2*, and *-3* (Figure 1A–D; Appendix A). The classification result from the analysis of motif composition was in agreement with the above *OsBAGs* family classification (Figure 1D). Of the total 10 motifs predicted, group I had eight conserved motifs, with the four in common shared by *OsBAG1*, *-2*, and *-3*; however, these proteins differed in the prediction of their localizations within cells (Table 1; Figure 1D).

### 3.3. Expression Patterns of OSBAG Genes in Rice

To observe the expression profiling of the *OsBAG* gene family, microarray datasets from 33 tissues, covering almost the whole life cycle of rice, were initially collected from CREP (http://crep.ncpgr.cn, accessed on 18 April 2021). Generally, *BAG* genes (*1*, *3*, and *4*) exhibited a high expression level in many tissues at different developmental stages of rice (Figure 2). *BAG4* showed the highest expression level among *BAGs* with the values ranged from 1299 to 6584 (average: 2962). *BAG1* showed the second highest expression level with the values ranged from 100 to 7048 (average: 2105). *BAG3* demonstrated the varying levels of expression, as extremely higher transcript levels were detected in radicle 48 h after emergence and in stem at the heading stage, whereas the transcript expression was not detectable in old flag leaf, sheath, and very young endosperms 2 and 3. We noticed the moderate expression of *BAG3* at the callus and panicle stages and in endosperm 1 with signal values of up to 300. Usually, the values above 50 were considered as the significant in the CREP experiment. Interestingly, the *BAG1* expression was much higher than the *BAG3* expression in much younger tissues, although these two genes had high similarities in expression pattern. This difference between *BAG1* and *BAG3* was also the case in a previous study [20]. We noticed that *BAG1* had higher expression than *BAG 3* in the very young tissues or organs including the callus, suspension cell, stigma, ovary, embryo, and endosperm. In contrast, *BAG6* only showed trace expression in most of the tissues examined, but preferentially high expression levels in several tissues including panicle, endosperms, and calli. Since *BAG2* and -*5* had no probe support in the CREP database, as an alternative, we searched the RiceXPro database for the expression data of these two genes. It was found that the expression pattern of *BAG2* was similar to those of *BAG1* and -*3*, while the expression level of *BAG5* was relatively low but was the highest in the ovary. In addition, we collected some tissues or organs at several stages as the representatives for qRT-PCR to verify the CREP expression data. It was revealed by the qRT-PCR that the expression level of *BAG1* was higher than that of *BAG3*, but the expression patterns were similar to each other. Thus, the expression levels of *BAG1* and -*3* were verified by qRT-PCR analysis (Appendix A).

### 3.4. OsBAG Co-Expression Profiling and Functional Relevance Analysis

Genes that are similarly expressed tend to be associated with the related biological process [39]. Thus, the co-expression analysis of *OsBAG* family genes was conducted. In this study, individual *BAG* genes were used as a bait to query the top 100 list of co-expressed genes using the CREP database. Then, co-expressed modules (BAG top 100 list) were subsequently interpreted by annotation and functional enrichment analysis. First, the top 100 genes that were tightly co-expressed with each *BAG* gene were identified by their transcriptome similarity throughout the developmental stages of the life cycle (Appendix A). We found that the top 100 co-expressed genes reflected biologically relevant information. Strikingly, in the top 100 co-expressed genes of *BAG1*, the genes for cell wall biosynthesis were over-represented. These over-represented genes included the relevant genes for typical polysaccharide biosynthesis such as xyloglucan endotransglycosylase/hydrolase protein 8 precursor (LOC_Os08g13920), beta-1,3-galactosyltransferase sqv-2 (LOC_Os02g06840), CSLF6—cellulose synthase-like family F beta1,3;1,4 glucan synthase (LOC_Os08g06380), sucrose synthase 2 (LOC_Os04g24430), glycosyltransferase (LOC_Os06g49320), fasciclin domain (LOC_Os07g06680), CESA8—cellulose synthase (LOC_Os07g10770), and COBRA-like protein 4 precursor (LOC_Os05g32110). Interestingly, similar to those of *BAG1*, the most of the top 100 co-expressed genes of *BAG3* were tightly correlated with cell wall biosynthesis, such as the genes of the fasciclin-like arabinogalactan (LOC_Os09g07350), CSLF6- cellulose synthase-like family F; beta1,3;1,4 glucan synthase (LOC_Os08g06380), CESA8—cellulose synthase (LOC_Os07g10770), CESA1-cellulose synthase (LOC_Os05g08370), and CESA3- cellulose synthase (LOC_Os07g24190). However, the top 100 co-expressed genes of *BAG4* were distinctive with those of *BAG1* and *-3* and showed that genes associated with phytohormone and transcriptional factors were significantly enriched such as the OsWRKY34-Superfamily TFs (LOC_Os02g43560), auxin-repressed protein (LOC_Os05g14180), and sIAA17—Auxin-responsive Aux/IAA gene family member (LOC_Os03g22270). Many of the top 100 co-expressed genes of *BAG6* were associated with disease and heat shock as represented by the members of NBS-LRR disease resistance protein (LOC_Os02g38386), heat shock factor protein HSF30 (LOC_Os10g28340), and heat shock factor protein 4 (LOC_Os04g48030) (Appendix A). To show the results more clearly, four co-express networks were constructed with the top 20 co-expressed genes of each *BAG* (selected from the top 100), which clearly displayed the tight co-expression relationships between the genes (Figure 3A). From the networks, the genes related to cell wall biosynthesis were obviously in the close vicinity of *BAG1* and *-3*, whereas the genes involved in the regulation of stress response were in the close vicinity of *BAG4* and *-6* (Figure 3A).

To assess the functional relevance of the genes in top 100 and to make sure that the co-expressed genes reflected biologically relevant information, we tested whether certain ontology terms were over-represented in the top 100 genes. GO enrichment analysis was therefore performed using a weighted method and Fisher’s exact test [40]. We furthermore performed the GO analysis of KEGG Orthology for each top 100 genes using hypergeometric tests [41]. Notably, a significant over-representation of the cellulose and non-cellulosic polysaccharide biosynthesis was observed for both the top 100 genes of *OsBAG1* and *-3* (Figure 3B). A significant over-representation of the genes for the nucleic acid metabolic process, the regulation of gene expression, and the Auxin-activated signaling pathway was observed for the top 100 genes of *OsBAG4*, while a similar result was found for the top 100 genes of *OsBAG6* with the MAPK cascade, transcription factors activity, actin filament bundle assembly, and the auxin-activated signaling pathway being over-represented.

To further demonstrate the possible roles of *OsBAGs* on the cell walls or/and plant development, a hierarchical cluster analysis was conducted (Figure 3B). It was shown that *BAG3* was strongly co-expressed with *OsCESA1*, *-3*, and *-8*, which are typical genes of forming a cellulose synthase complex for primary cell wall biosynthesis as reported in our previous study [28]. The expression of this gene group was highlighted in the young tissues undergoing both primary and secondary growth, including the seed inhibition, plumule and radicle after germination, the young shoot and root at the seedling stage, and the young stem at heading stages. However, *BAG1* was clustered with *OsCESA2*, *-5,* and *-6*, which are preferred for non-cellulosic polysaccharide, instead of cellulose, as reported in our previous studies [28,39]. This group of genes showed higher expression, not only in the tissues similar to the *BAG3* group, but also in the calli, panicle, endosperms, and the younger tissues mainly experiencing primary growth. The slight difference in the cluster infers the differences in the function of *BAG1* and *BAG3*, despite the fact that both genes are predicted to be involved in cell wall biosynthesis. By contrast, *BAG4* and *BAG6* did not show obvious co-expression with the genes encoding cellulose synthase. However, it should be mentioned that *BAG6* prefers to be co-expressed with *CesA11* and with some genes involved in actin filament bundle assembly, possibly indicating some functional overlaps with the cell wall modification process.

### 3.5. OsBAG3 Exhibited Young Tissues-Preferential GUS Staining Patterns, Consisting with the Transcriptional Profiling

To further confirm the expression of *OsBAGs* in rice plants, the vector containing GUS reporter gene driven by *OsBAG3* promoter (*pOsBAG3::GUS*) was constructed and introduced into the rice variety Nipponbare. The histochemical staining of the GUS activity in transgenic lines revealed that the *BAG3* gene was widely expressed in most tissue examined across the entire life circle, basically matching the expression pattern of the gene obtained from the CREP database. GUS signals were stronger detected in calli, the coleoptile of germinated seeds, plumule and radicle after germination, the leaves at the seedling stage and the tilling stage, the stem, sheath, hull and pedicel at the booting stage, and the stem and hull at the heading stage (Figure 4). Although the GUS activity was also detected in the tissues or organs under the secondary growth, it is likely that *BAG3* was mainly involved in primary cell wall biosynthesis in both the primary- and secondary-growth tissues inferred from the following four points. Firstly, it showed that in general the GUS staining was much stronger in the tissues at young stages than those at old or at mature stages. Secondly, in the same tissue or organ, the histochemical staining was much stronger in the early phase than that in later phase. For example, the GUS signals became weaker in plumule during germination, and the GUS signals were much stronger in leaves at the seedling stage than those at the tilling stage (Figure 4A,B). GUS signals were observed in both hull and the pedicel at the booting stage compared to the weak or no staining in those tissues at the heading stage (Figure 4C,D). Thirdly, the tendency was observed at the same tissues or organs in different development situations at a stage. At the booting stage, we collected the leave, sheath, internode, and node with the collar (auricle) from the different sections of the stem and found that the GUS staining in these tissues became weaker from the immature (inverted 1st internode) to the intermediate (inverted 2nd internode) and was not detectable at the mature (inverted 3rd internode) (Figure 4C). At the heading stage, three different fragments (upper, middle, and bottom) of the uppermost internode were cut and separately stained, and it was observed that the lower internode near the intermediate meristem was much stronger than the others (Figure 4D). Fourthly, of the leaves at any development stages, less or no GUS staining was found in the vein tissues containing the vascular bundles where the primary cell wall biosynthesis ceased, whereas in the nonvein areas the GUS staining was strong. Based on the above observations, we identified *BAG3* exhibited young tissues-preferential GUS staining patterns, consistent with the results revealed by the microarray data analysis, very similar to the expression pattern of cellulose synthases responsible for the synthesis of primary cell walls.

It has to be mentioned that, although the histochemical staining of the GUS activity in *pOsBAG3::GUS* transgenic lines basically matched the expression pattern of the gene obtained from the CREP database, there was a little inconsistence between two results. It seems that a strong GUS signal in the calli revealed by staining analysis was not consistent with the moderate expression of *BAG3* in the calli in Figure 2. One of the reasons might be the staining problem caused by the excessive adsorption of the X-Gluc in the calli. On the other hand, the degree of GUS staining in the leaves could be weaken by the interference of chlorophyll, which may not be thoroughly eliminated by washing buffer. In addition, in the case of the strong GUS staining, the linear relationship between the staining intensity and the gene expression did not work well; it is difficult to draw an accurate conclusion in this case. These uncontrolled reasons will contribute to the inconsistence between the GUS staining and the gene expression level.

### 3.6. Expression Profiles of OsBAGs in Responses to Environmental Stimulations

Since the *AtBAGs* of the *Arabidopsis* were reported to be involved in responses to many stimulations, it is also of interesting to investigate the *OsBAGs* expression profiles in response to environmental stimuli. We first fed the rice plants with BPH at seedling stages and to examine whether *OsBAGs* responded to an attack by the BPH. After the BPH treatment, the expression levels of *OsBAG1*, *-2*, and *-3* were decreased at 24 h and were much significant at 48 h, whereas those of *OsBAG4, -5*, and *-6* were unchanged after the BPH treatment (Figure 5A).

Nitrogen (N), phosphorus (P), and potassium (K) were the macronutrients essential for plants. It was shown that the expression levels of all *OsBAG* genes were significantly increased at time points of 6–12 h under the conditions deficient in N, P, and K, compared with that of the control. Especially, under the N deficient condition, the expression of all *BAG* genes increased rapidly at the early time point of 6 h (Figure 5B). We further investigated the expression of the *OsBAGs* under the stresses of heat and drought using the data downloaded from the NCBI GEO database. It was found that after continuous high temperature treatment, *BAG1* and *-3* were upregulated (at least four-fold), while *BAG4* was repressed. The expression of *BAG6* was found to be induced in the heat-susceptible line 082 but remained unchanged in the heat-tolerant line 252 (Figure 5C). Drought or osmotic stress is one of the major abiotic stresses afflicting crop plants. It was shown that *BAG1* and *-3* were repressed in two contrasting varieties Dagad deshi and IR20 [32,33]. *BAG4* was not drought-responsive, while the expression of *BAG6* was increased in drought-tolerant line Dagad deshi but remained unchanged in the susceptible line IR20 (Figure 5D). Interestingly, the expression of *BAG1* and *-3* was co-regulated in responses to environmental stresses.

The growth-promoting hormones such as IAA and gibberellin 3 (GA_3_) regulate diverse developmental processes throughout the life cycle of the plants, while salicylic acid (SA) and JA are well-known signaling molecules that mediate plant defense response [42]. We found that the expression of *BAG1* was induced in shoots but repressed in roots after the treatments of ABA and IAA (Figure 6). *BAG3* was again similar to *BAG1* in the expression (Figure 6). However, its expression was induced after the treatment of the MeJA in both shoots and roots, which was different with that of *BAG1*. However, the expression of *BAG2* differed from those of *BAG1* and *-3,* which was only found to change in the roots, with the level decreased after the treatments with ABA and IAA and the level increased after Zeatin (auxin) treatment (Figure 6). The expression of *BAG4* was only induced in the roots after the treatments with ABA and MeJA (Figure 6). The expression profile of *BAG5* seemed the combination of those of *BAG2* and *-3*. In particular, *BAG6* responded strongly to all the treatments in the early time (Figure 6). In conclusion, *BAG* genes were mostly associated with ABA and MeJA, but they showed fewer changes in their transcripts after the BL treatment. *BAG5* and *-6* were more sensitive to the hormone treatments than the other *BAGs,* consisting with the findings that they have the highest numbers of the responsive elements for ABA, MeJA, and light treatments within their promoter regions (Appendix A). Once again, *BAG1* and *-3* were found to be co-regulated in the responses to phytohormone treatments. 

An overview of the expression profile of *OsBAG* genes in root and shoot from rice seedlings treated with six plant hormones, namely ABA, GA3, IAA, BL, trans-zeatin (tZ), and JA. The chart was analyzed using the ratio of the RAW signal intensities of treatment (Cy5) and control (Cy3).

## 4. Discussion

The BAG proteins are a multifunctional group of chaperone regulators, which are reported to participate in several cellular events, including apoptosis, proliferation, differentiation, and stress signaling in animals [2,11,20,43,44,45,46]. Most studies reported that *BAGs* regulate processes from pathogen attack to abiotic stress such as cold, drought, and salt and are involved in cellular responses based on the Ca^2+^/CaM signaling pathway [11]. However, less work has been conducted to demonstrate the influence of *BAGs* on plant growth and development [11,47].

This study identified six isoforms of OsBAG proteins in rice. Among these, BAG1–4 comprise a subfamily of proteins that are predicted to possess UBL domains and BDs. BAG5 and -6 comprise another subfamily containing a unique IQ domain (for calmodulin binding) as well as a BD [2]. Then, we performed an integrated analysis of genome organization, subcellular localization, molecular phylogeny, and protein structure to provide the basis for further functional studies. In addition, fortunately, rice has a wealth of global gene expression databases and several co-expression networks in RiceArrayNet (PlantArraynet) [48,49]. Among them, the CREP database and the RiceXPro database are of high quality, as they contain the genome-wide expression data covering a large number of heterogeneity samples across the whole life cycle of rice, which allows the increased power of the co-expression analyses at the same time and decreased “positive-false” possibility [28,32,33,39,50,51]. Thus, we can accomplish a comprehensive understanding of the functions of the rice *BAG* genes by integrated bioinformatic analysis, and our method is effective in identifying and ranking the over-represented functional categories of *BAG* genes.

### 4.1. The Expression Levels of OsBAG1 and -3 Were in High Similarity but Subtle Differences during the Plant Development

Some previous studies indicated the association of a small number of *BAGs* with plant development; however, the mechanism underlying remains to be elucidated. *BAG1* and *-3* are co-regulated in stress response, including BPH, N, P, and K deficiency, high-temperature treatment, drought treatment, and the hormone treatments. Their expression patterns tend to be consistent with each other, indicating the close similarity between *BAG1* and *BAG3* and the possible redundancy of their functions. Furthermore, one of the striking findings in our study was the tight association of *BAG1* and *BAG3* with the cell wall biosynthesis genes by co-expression analysis and the GUS staining. Firstly, *BAG1* was tightly co-expressed with the genes involved in the synthesis of non-cellulosic polysaccharides including xyloglucan endotransglycosylase (top 7), beta-1,3-galactosyltransferase (top 17), CSLF6 beta1,3;1,4 glucan synthase (top 19), and glycosyltransferase (top 29), whereas the *BAG3* was tightly co-expressed with the genes that were assigned to both cellulosic and non-cellulosic polysaccharides including the CSLF6 beta1,3;1,4 glucan synthase (top 3), CESA8 cellulose synthases 8 (top 3), beta-1,3-galactosyltransferase (top 16), and CESA1 cellulose synthases 1 (top 20). The result was subsequently strengthened by an integrated analysis of co-expression network, GO enrichment, and KEGG. Secondly, the hierarchical cluster analysis clearly showed that *BAG3* was grouped with three cellulose synthase genes *OsCesA1, -3*, and *-8* typically for primary cell wall cellulose biosynthesis, whereas *BAG1* was clearly grouped into another distinct group (Appendix A). We further calculated the pairwise correlation coefficient between the *CesAs* and *BAGs* and found that the primary wall cellulose biosynthesis genes (*CesA1*, *-3*, and *-6*) showed much strong positive correlation with *BAG3* (r values from 0.8 to 0.9) than with that with *BAG1* (r values from 0.6 to 0.7) (Appendix A). Furthermore, compared to *BAG3*, *BAG1* also showed considerable expression in the calli, younger panicle, and endosperms, the tissues with a considerable amount of non-cellulosic polysaccharides (hemicellulose and pectin) mainly under primary growth. The difference between the two genes was also the case in a previous study reported by Rana et al. [20], in which we noticed that *BAG1* has higher expression than *BAG3* in very young tissues or organs including the callus, suspension cell, stigma, ovary, embryo, and endosperm. Taken together, it is clear that *BAG 1* and *-3* were highly coordinated with the primary cell wall biosynthesis, indicating that they are associated with cell growth and development and possibly are active in the same biological process during cell wall formation. In detail, we suspected that *BAG1* was more likely associated with non-cellulosic polysaccharides throughout the whole life cycle, whereas the *BAG3* was relevant to both cellulosic and non-cellulosic polysaccharides in both primary growth and secondary growth. Many genes that affect polysaccharides or lignin deposition, including COBRA and CTL1/POM1, are co-expressed with the CESA genes [28,39]. The successful cloning of some cell wall-related genes such as the “cellulose synthase-interactive protein 1” CSI1 have given biologists many expectations [52]. *CSI1* was identified previously as one of the genes that is co-regulated transcriptionally with the primary *CesAs*, and finally, its biological role in the cell wall biosynthesis was confirmed [52,53,54]. Our results suggested, for the first time, potential roles of *BAG1* and *BAG3* associated with the primary cell wall biosynthesis. However, since the co-expression results are considered just as indicative, further researches are required to confirm their potential function in cell wall biosynthesis plant development.

### 4.2. The Divergent Roles of the Rice BAG Genes in Plant Development and in Responses to Environmental Stimulations

Based on the analysis of the *BAGs* with an unrooted phylogenetic tree, exon–intron organization, and motif composition, the six rice BAGs proteins could be divided into two groups, group I (BAG1, -2, -3, and -4) has the ubiquitin domains and BDs and group II (BAG5 and -6) has the IQ domains and BDs, which could possibly reflect the functions of BAG proteins. The multifunction of the BAGs was also supported by the predicted subcellular localization of the BAGs, as BAG1, -2, and -3 were mainly localized in the cytosol with the possibility in nucleus and plastid while BAG4 and BAG6 were located in cytosol and nucleus separately. In addition, BAG1, -2, and -3 in group I had a high similarity in sequences and four motifs in common, and they differed slightly in the prediction of the protein localization within cells. Further studies are needed to determine their subcellular localization by using the GFP-tagged BAG proteins. Co-expression analysis has been proved successful in revealing functional relationships and common biological pathways between the gene products across many species [39,53,55,56,57]. In *Arabidopsis*, the BAG proteins were reported to regulate apoptosis-like processes ranging from pathogen attack and development to abiotic stresses [11]. In this study, we clearly showed that the expression patterns of the *OsBAG* genes were different with each other. Basically, the family of the *OsBAG* is classified into two groups. It seems that the genes of group I were mostly involved into the plant development, while those in group II were possibly active in gene regulation and stress responses. However, the members within a group also have some differences in structure and expression pattern. A case in point is that *OsBAG1* and *-3* also had some subtle difference as indicated by thorough co-expression analysis. According to their genome organization, protein structure, and the expression patterns across the development and the responses to the environmental stimulates, our results implied that the rice BAG family members are multifunctional in development and stresses.

## Figures and Tables

**Figure 1 plants-10-02169-f001:**
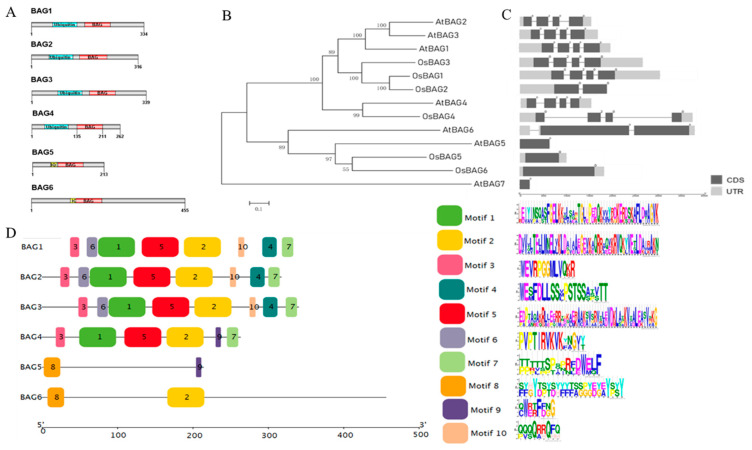
Structural and phylogenetic analyses of *OsBAGs.* (**A**) Domain structures of the OsBAG proteins. The positions of the BAG domain (BD; red), ubiquitin-like domain (blue), and calmodulin-binding motif (yellow) are shown. (**B**) Phylogenetic relationships of 13 *OsBAG*/*AtBAG* proteins. The phylogenetic tree was constructed with MEGA 6.0 software using the neighbor-joining (NJ) method with 1000 bootstrap replicates. (**C**) Gene structures (exon–intron organization) analysis of *OsBAG*/*AtBAG*. The gene structures were drawn online with Gene Structure Display Server 2.0. The CDSs, introns, and UTRs are marked with black boxes, gray lines, and gray boxes, respectively. The scale bar is shown at the bottom. (**D**) Analysis of the conserved domains of the OsBAG proteins. Differently colored boxes represent different conserved motifs of OsBAG proteins.

**Figure 2 plants-10-02169-f002:**
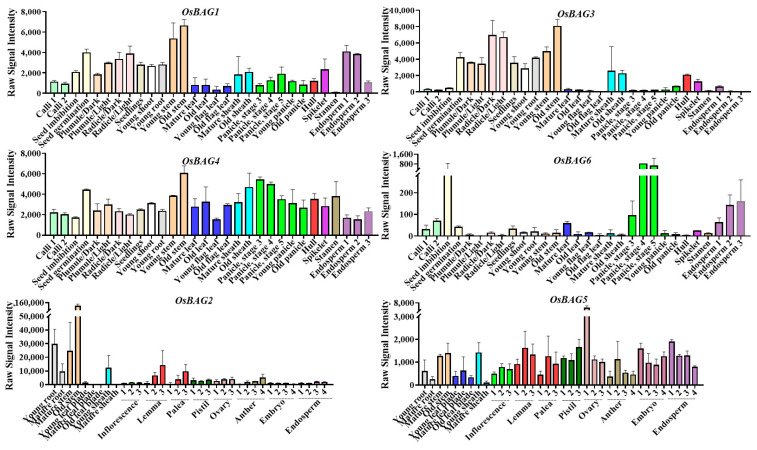
The expression patterns of *OsBAGs* at different developmental stages of rice. The expression patterns of the *OsBAG1*, *-3*, *-4*, and *-6* in the varieties Minghui63/Zhenshan97 were based on Collections of Rice Expression Profiling (CREP), and those of *OsBAG2* and *-5* in the Nipponbare were based on RiceXPro data.

**Figure 3 plants-10-02169-f003:**
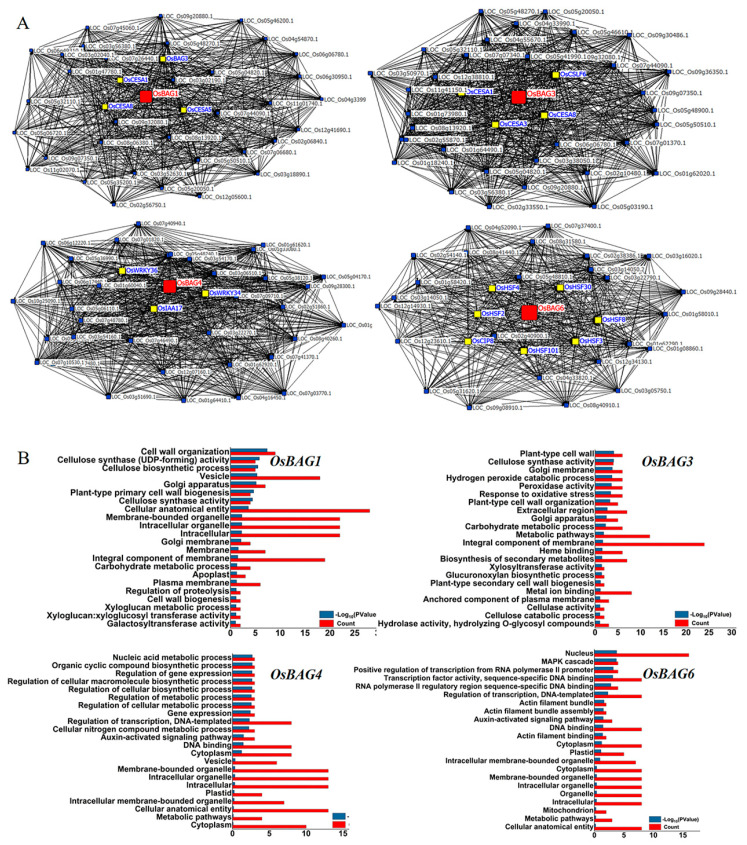
*OsBAG* co-expression profiling and the functional relevance analysis. (**A**) Co-expression gene vicinity network for *OsBAGs*. Four co-express networks were constructed with the top 20 co-expressed genes of each *OsBAG* (selected from the top 100), which clearly displayed the tight co-expression relationships between the genes. (**B**) Gene Ontology (GO) analysis of *OsBAG* genes.

**Figure 4 plants-10-02169-f004:**
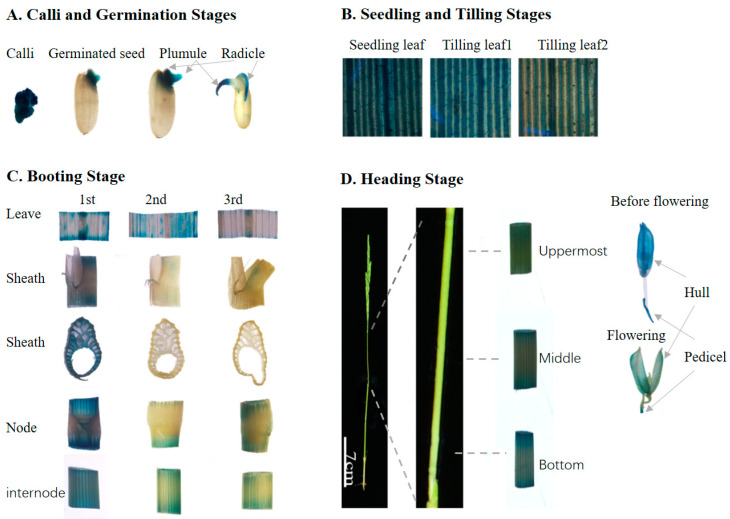
Analysis of β-glucuronidase (GUS) activity in the *pOsBAG3::GUS* transgenic plants. The GUS activity is shown in most tissue examined across the entire life circle. 1st, inverted 1st internode; 2nd, inverted 2nd internode; 3rd, inverted 3rd internode; Uppermost, upper fragments of the uppermost internode; Middle, middle fragments of the uppermost internode; Bottom, bottom fragments of the uppermost internode.

**Figure 5 plants-10-02169-f005:**
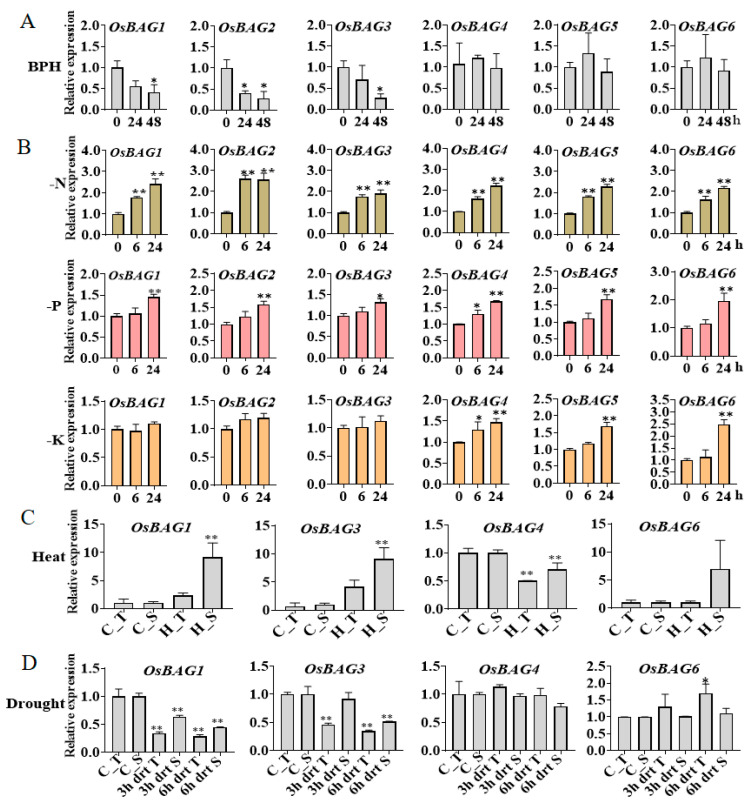
Expression patterns of *OsBAGs* in responses to environmental stimulations. (**A**) Gene expression profiles of rice in response to brown planthopper (BPH). The expression patterns of *OsBAG* genes were analyzed by real-time qPCR. The 14-day-old rice seedlings were treated with the BPH feeding from 0 to 48 h. *n* = 5. (**B**) An overview of the expression profiles of all genes in root from rice seedlings under nitrogen (N), phosphorus (P), and potassium (K) deficiency conditions based on the RiceXPro data. The 7-day-old seedlings were exposed to N, P, and K deficiency treatments and control conditions separately. (**C**) Expression data collected from the panicles of both heat-resistant and heat-sensitive rice under high temperature. C_T, control tolerant; C_S, control sensitive; H_T, heat tolerant; H_S, heat sensitive. (**D**) Hydroponically grown seven-day-old seedlings of drought tolerant and susceptible cultivars of *indica* were subjected to drought stress. C_T, control-tolerant; C_S, control-sensitive; 3 h drt T, 3 h drought-tolerant; 3 h drt S, 3 h drought-sensitive; 6 h drt T, 6 h drought-tolerant; 6 h drt S, 6 h drought-sensitive. The expression data in (**B**–**D**) based on NCBI GEO DataSets. * and ** indicate the significance at *p* = 0.05 and *p* = 0.01 levels by Student’s *t* test, respectively, and bars represent standard deviations.

**Figure 6 plants-10-02169-f006:**
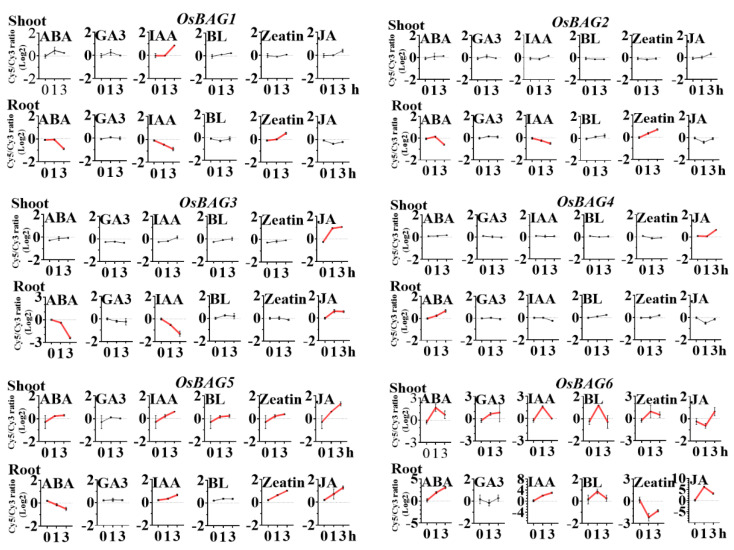
Expression profiles of *OsBAGs* in responses to plant hormones.

**Table 1 plants-10-02169-t001:** List of the six *OsBAG* genes identified in rice.

Genes	TIGR Loci	Probsets	CDS/bp	KOME cDNA	Exons			Protein		
						Length (aa)	PI	Domain	Molecular Weight (kDa)	Predicted Location (s)
*OsBAG1*	*LOC_Os09g35630.1*	Os.51065.1.S1_at	1005	AK105833	4	335	9.65	Ubiquitin (PS50053), BAG (PS51035, PF02179)	35.96	Mitochondrion.
*OsBAG2*	*LOC_Os08g43270.1*	/	951	FP095381	2	317	9.61	Ubiquitin (PS50053), BAG (PS51035, PF02179)	34.7	Cytosol.
*OsBAG3*	*LOC_Os06g03640.1*	Os.10179.1.S1_at	1020	AK065197	4	340	9.71	Ubiquitin (PS50053), BAG (PS51035, PF02179)	36.43	Nucleus.
*OsBAG4*	*LOC_Os01g61500.1*	Os.20681.2.A1_at	789	AK070208	4	263	5.63	Ubiquitin (PS50053), BAG (PS51035, PF02179)	28.78	Cytosol
*OsBAG5*	*LOC_Os02g48780.1*	/	642	AK119930	1	214	5.99	IQ (PS50096), BAG (PS51035, PF02179)	23.08	Cytosol.
*OsBAG6*	*LOC_Os11g31060.1*	OsAffx.19095.1.S1_at	1368	FP100206	1	456	4.48	IQ (PS50096), BAG (PS51035, PF02179)	44.3	Nucleus.

## Data Availability

Not applicable.

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
