# Peer review of "The Divergent Roles of the Rice bcl-2 Associated Athanogene (BAG) Genes in Plant Development and Environmental Responses"

_plants, 2021, doi:10.3390/plants10102169_

Round 1
Reviewer 1 Report
Research presented by Authors is generally important, novel and original.
However, the materials and methods section as well as results presentation should be improved.
Below are precise points to be used in manuscript correction:
Section 2.4
Authors decided to use correlation coefficients (r) higher than 0.6.
It is generally ok, and related to data presented by Usadel et al. 2009 (Co-expression tools for plants biology: opportunities for hypothesis generation and caveats) suggesting to use values r even higher than 0.7 to assure that obtained results are biologically relevant. Authors may cite this work or related research to support their choice of r value.
Section 2.5
More precise description of RT-PCR experiments is necessary, following points could be added:
Assessment of RNA purity, concentration and integrity
Details of DNase treatment to remove putative remnants of genomic DNA
Reverse transcription reaction; details of reaction-temperature and time, volume and amount of used RNA
qPCR target- length of PCR product (control and tested gene), target gene symbol and accession number,
qPCR protocol: conditions of PCR reaction, volume of reaction, concentration of magnesium ions, dNTPs, DNA polymerase type and concentration, additives as SYBR Green, DMSO etc. Name and manufacturer of qPCR instrument.
Justification of RT-PCR reference gene choice- length of reference PCR product length
Software used to analyse RT-PCR results, statistical method.
Citation of method used to calculate RT-PCR results- for example Livak and Schmittgen 2001 or other.
Section 3.1
Authors write OsBAG 3 & 6- it should be OsBAG3 and 6 or OsBAG3,6. The symbol & should be removed from the text- the same for the abstract and the whole manuscript.
Section 3.3
Units of expression level (RPM, RPKM, FPKM, TMM, TPM etc) should be provided-also in Fig 2.
Section 3.4
Ranges of r values for top co-expressed genes should be provided.
Section 2.7
Putative mistake in isolated promoter length, Authors wrote that it was 11731bp, however, plant promoters usually have about 1kb (about ten times shorter as provided) and most of functional cis-active sites are within proximal promoter and 5’UTR. The 11731bp should be corrected.
Details of PCR reaction used to isolate the gene promoter should be provided, method of isolation of DNA template (putatively genomic DNA) should be presented.
Details of transcriptomic experiments using Affymetrix Gene Chip should be provided:
Experiment design is described relatively good but there are lacks in other parts, for example:
Array type- for example glass spotted
Description of the whole array – platform type, provider, surface type
Description of each element of spot- for example- synthesized oligo nucleotides or PCR products from cDNA library
Information of DNA sequences used in a spots
Samples preparing- method of nucleic acid extraction and labeling
Details of hybridization reaction- composition of hybridization solution, blocking agent, hybridization time and volume, wash method, hybridization instruments-manufacturer, name of instrument.
Description how the progress from raw to final microarray data was done-applied software.
Methods used to normalize microarray results- for example based on total intensity, ratio-based , linear or nonlinear regression or other.
Other:
Latin names of plants and animals should be written in italic-whole text.
Software and method used to construct networks of co-expressed genes should be presented.
It is not clear if data in Fig 6 are transcriptomic data obtained in Microarray experiments performed by Authors or are obtained from a database. Please describe precisely in Results and Material and methods sections.
Reviewer 2 Report
Zhou and collaborators, in the manuscript entitled “The divergent roles of the rice bcl-2 associated athanogene (BAG) genes in plant development and environmental responses” have done an interesting overview of the BAG genes expression profiles and response to different environmental stresses and hormonal response.
However, some of the experiments presented here were largely confirmative and one of the manuscript already published described some of the results presented here. (Rana et al., 2011).
Results:
Table 1:
The predicted localization of the BAG proteins was already described in Rana et al., 2011. Rana and co-workers predicted that OsBAG1 is localized in the chloroplast and OsBAG6 in the cytosol. Instead, the authors in this manuscript described that OsBAG1 is located in the mitochondrion and BAG6 in the Nucleus.
Why differences like these are not discussed in the text? Why the prediction of the authors is different respect to that already described?
Figure1 (Structural and phylogenetic analyses of OsBAGs)
The phylogenetic relationship between Arabidopsis and Oryza sativa BAGs was already well described by Rana et al., 2011. Moreover, this Figure, which described also why the OsBAGs were divided in two groups, confirmed data that were already proposed in literature.
Figure 2 and Additional File 6 (The Expression patterns of OsBAGs at different developmental stages)
Which are the differences between the two Figures?
Figure 2
I agree with the authors that in some points BAG1 and 3 have some similarities in its expression. But it is clear, from the data represented in the Figure, that BAG1 is expressed in the Callus while BAG3 is not expressed. Moreover, BAG3 is not expressed in the Panicle stages and in Endosperm 2 and 3 while BAG1 has a higher expression in these stages.
How do the authors explain these differences between the two genes, that you considering co-expressed and with the same functions during plant development?
I would like to see a confirmation of these bioinformatics data, described in the Figure 2, with qRT-PCR in order to see if the similarities among the various genes will be confirmed.
Figure 3
In my opinion the correlations among the BAGs genes and CESA or WRKY genes, revealed by co-expression analysis could be just an indication of BAGs activities in plants;
It is not supported by this experiment the conclusion described in the discussion:
“Our results for the first time suggested a potential role of the BAG1 and BAG33 associated with the primary cell wall biosynthesis”;
In my opinion, they should need a bag1 or bag3 mutant line, in which there will be less/high transcript levels of the CESA genes described, and also an increase/decrease of cellulose in the primary cell wall.
Figure 4
The Figure was nice arranged. However, in the Figure4, OsBAG3 pro::GUS analysis revealed a strong GUS signal in the Calli, while the expression pattern in the Figure2 of the same gene revealed that BAG3 was not expressed in the Calli1, and there was a low expression in Calli2.
How do you explain these differences?
For this reason, it could be better a qRT-PCR of BAG3 in the stages analysed with the GUS assay.
Why did the authors choose only BAG3 for the GUS assay? It should be better, in order to confirm the co-expression pattern with BAG3, also to perform the same experiment on BAG1.
Figure 5
It should separate in this Figure what is an experiment that has done in the lab by the authors, and what was obtained by the analysis of different dataset. *, ** Missing information, this is a Statistical test?
Minor point:
“The growth-promoting hormones such as indole-3-acetic acid (IAA) and gibberellin 3 (GA3) regulate diverse developmental processes throughout the life cycle of the plants, while the salicylic acid (SA) and jasmonic acid (JA) are well-known signaling molecules that mediate plant defense response”.
At the end of this paragraph, the authors should be put the references.
Figure 6
The Figure is full of data, but it is small.
Reviewer 3 Report
The authors, in this manuscript describe the rice BAG family of genes and provide some insights in their biological roles.
The manuscript is well presented and well written in clear way.
The data provided support their claims and conclusions.
Some minor issues:
-Fig. 3 is hard to read, needs better quality of pictures and/or bigger letters
-Fig4, the GUS staining of the calli tissue is strong suggesting a ligh expression level but the expression data at Fig 2 show a very low expression, could you provide an explaination for this?
-Fig5 maybe could you reduce the text, especial for C and D sections, for example the control tolerant variety write as C-T
-Fig 6 too many data, hard to read, maybe the authors could show only the data that change significantly and provide the rest as supplementary or use a heatmap or other visualization methods
Round 2
Reviewer 1 Report
Authors significantly improved manuscript according to suggestions.
Only minor corrections should be introduced:
Paragraph 2.4
tissue examples- putatively better could be tissue samples
Paragraph 2.5
Analytikjena- should be Analytik Jena
Authors should provide equipment used to determine the RNA amount used in RT-PCR analysis- usually it is done by microvolume nanophotometers for example produced by Implen or other company. Determination of starting RNA amount is crucial for RT-PCR analysis.
Also the length in bp of tested (OsBAG1-6) and reference gene (Osactin) PCR product used in RT-PCR analysis together with their accession numbers should be presented. It could be easily calculated by primer alignment with DNA/cDNA matrice sequence.
Authors should provide the citation of 2-DeltaDeltaC(T) method of Livak and Schmittgen 2001:
Analysis of relative gene expression data using real-time quantitative PCR and the 2(-Delta Delta C(T)) Method - PubMed (nih.gov)
Other comments:
Authors write reference (entire Author names) nr 29 in capital letters- write it as other references
Reviewer 2 Report
The authors have answered to most of the criticisms of the manuscript; I suggest some minor changes in order to improve the manuscript;
- For the Additional file 7, the authors should do a statistical analysis like t-test in order to confirm that the expressions among the two BAGs are different in some developmental stages.
“It has to be mentioned that, although the histochemical staining of GUS activity in pOsBAG3::GUS transgenic lines basically matched the expression pattern of the gene ob-tained from the CREP database, there is a little inconsistent between two results….
- Please could you write in italics form pOsBAG3::GUS.
- In the same paragraph please add a space “BAG3 in calli in the Figure2”
Figure 4. Examination of GUS activity in the transgenic plants expressing OsBAG3 pro::GUS.
- I think that you should change the title of the Figure in:
Analysis of GUS activity in the pOsBAG3::GUS transgenic plants.
Figure 5 “The expression patterns of OsBAG genes were analyzed by real-time qPCR.”
- Please change real-time into Real-Time qPCR or qRT-PCR, also change it into Materials and methods.
Author Response
Please see the attachment.

This manuscript is a resubmission of an earlier submission. The following is a list of the peer review reports and author responses from that submission.